# Infection, Transmission, Pathogenesis and Vaccine Development against *Mycoplasma gallisepticum*

**DOI:** 10.3390/vaccines11020469

**Published:** 2023-02-17

**Authors:** Susithra Priyadarshni Mugunthan, Ganapathy Kannan, Harish Mani Chandra, Biswaranjan Paital

**Affiliations:** 1Department of Biotechnology, Thiruvalluvar University, Vellore 632115, India; 2Institute of Infection, Veterinary & Ecology Sciences (IVES), University of Liverpool, Neston, Cheshire CH64 7TE, UK; 3Redox Regulation Laboratory, Department of Zoology, College of Basic Science and Humanities, Odisha University of Agriculture and Technology, Bhubaneswar 751003, India

**Keywords:** *Mycoplasma gallisepticum*, virulence factors, cytoadhesion, immune evasion, vaccine development

## Abstract

*Mycoplasma* sp. comprises cell wall-less bacteria with reduced genome size and can infect mammals, reptiles, birds, and plants. Avian mycoplasmosis, particularly in chickens, is primarily caused by *Mycoplasma gallisepticum* (MG) and *Mycoplasma synoviae*. It causes infection and pathology mainly in the respiratory, reproductive, and musculoskeletal systems. MG is the most widely distributed pathogenic avian mycoplasma with a wide range of host susceptibility and virulence. MG is transmitted both by horizontal and vertical routes. MG infection induces innate, cellular, mucosal, and adaptive immune responses in the host. Macrophages aid in phagocytosis and clearance, and B and T cells play critical roles in the clearance and prevention of MG. The virulent factors of MG are adhesion proteins, lipoproteins, heat shock proteins, and antigenic variation proteins, all of which play pivotal roles in host cell entry and pathogenesis. Prevention of MG relies on farm and flock biosecurity, management strategies, early diagnosis, use of antimicrobials, and vaccination. This review summarizes the vital pathogenic mechanisms underlying MG infection and recapitulates the virulence factors of MG–host cell adhesion, antigenic variation, nutrient transport, and immune evasion. The review also highlights the limitations of current vaccines and the development of innovative future vaccines against MG.

## 1. Introduction

The genus *Mycoplasma* is the simplest self-replicating microorganism, with genome size varying from 500 to 1500 kilobase pairs (kbp) [1]. Mycoplasmas are one of the most significant pathogens of poultry, goats, cattle, pigs, fishes, and a broad spectrum of other animals. Although mycoplasmas have small genome sizes, they employ complex molecular mechanisms for host-cell interaction and pathogenesis [1]. They are cell wall-less microbes phylogenetically related to low G+C Gram-positive bacteria. They followed degenerative evolution and belong to the class Mollicutes (In Latin: Soft skin) [1]. Mycoplasmas lack peptidoglycan cell walls and are surrounded by a single cytoplasmic membrane with lipoproteins. Mycoplasmas lack biosynthetic pathways and thus forage on their host for their nutritional necessities [2]. Mycoplasma lipoproteins play a critical role in host-cell interactions and pathogenesis since they commonly come into interaction with the extracellular environment [1].

In 1898, Nocard and Roux cultivated a bovine pleuropneumoniae-like organism, which led to the emergence of a new area of science—mycoplasmology. Till the early 20th century, there was a debate about whether mycoplasma was a bacteria or a virus. In 1935, a group of scientists identified that mycoplasma shares similar characteristics with L forms of bacteria (strains of bacteria that lack cell walls) in its filterability, morphology, and colony formation [3]. Later, for about 30 years, mycoplasmas were believed to be L-form variants of other common bacteria. With the advancement in genomic data analysis, the low guanine and cytosine content of mycoplasmas were estimated. Along with this, the inability to perform biosynthetic pathways and reduced genome size differentiated mycoplasmas from the bacterial L form [1,4]. Another unique characteristic of mycoplasmas is the use of the UGA codon to encode tryptophan, and this is due to the presence of tRNA, which is capable of reading the UGA termination codon as tryptophan [4].

A range of non-pathogenic and pathogenic mycoplasmas have been reported in different species of birds [5]. Around 23 species of mycoplasmas are been reported in birds. Among them, the most important, common, and pathogenic mycoplasmas are *M. gallisepticum*, *M. synoviae*, *M. iowae*, and *M. melegreditis* [5,6]. In chickens and turkeys, *M. synoviae* causes infectious synovitis mainly involving the synovial membranes of joints and tendon sheaths. In the last decade, *M. synoviae*, either as a single or mixed infection with infectious bronchitis and/or Newcastle disease viruses, has been reported to cause respiratory disease. *Mycoplasma iowae* infection results in reduced hatchability and embryo mortality along with leg troubles in juvenile turkeys [6]. *Mycoplasma meleagridis* is specific to turkeys, and its infection is associated with skeletal abnormalities and airsacculitis [6].

Out of the four mycoplasmas, *M. gallisepticum* (MG) is a major pathogen of gallinaceous and certain non-gallinaceous avian species [7], Figure 1. This highly transmissible organism is the etiologic agent of ‘chronic respiratory disease’ in chickens and other birds. Clinical symptoms of MG infections in these species include rales, coughing, sneezing, nasal discharge, and swollen infraorbital sinuses. The consequences of MG infection include mortality; increased carcass condemnation; reduced egg production, hatchability, and feed efficiency; and weight gain [8]. Due to the substantial performance and production losses, MG has been described as the most economically important pathogenic *Mycoplasma* species affecting poultry [8]. MG infection is reported worldwide and listed in the World Organization for Animal Health (OIE) [9].

MG can cause illness in a range of avian species, including chickens. MG causes infectious sinusitis in turkeys [10] and epidemic conjunctivitis in house finches in North America [11,12]. Later, MG was also reported to cause disease in songbirds, including American goldfinches [12], purple finches [13], and house sparrows [14]. Game birds, including pheasants and partridges, as well as chukar partridges, have been reported to be infected by MG [15]. It also causes infection in other birds including peafowl [16], Japanese quail [17], bobwhite quail, pigeons, ducks and geese [18], yellow-naped Amazon parrots, greater flamingos, and peregrine falcons [19,20]. MG in chickens not only causes CRD but also opens the way for other co-infecting pathogens, such as *E. coli*, and low pathogenic avian influenza subtype H9N1, causing severe economic losses [21].

Owing to the high pathogenic activity of MG in avian species that leads to a huge economic loss in avian industries, a comprehensive review is undertaken in the current article on this species covering its mode of infection, pathogenesis, and vaccine development status.

## 2. Transmission

*M. gallisepticum* follows two routes for transmission—either by transovarian (vertical) transmission or by lateral entry through direct contact (horizontal transmission) [21].

### 2.1. Vertical

Vertical transmission may happen in ovo (egg) and in embryo, and transmission to the progeny (egg transmission) possibly happens as a consequence of respiratory infection of hens, owing to the adjacency of the abdominal air sacs to the oviduct [22]. The transmission rates are expected to differ broadly under different conditions, among individual birds, and at different times during the same outbreak. The transmission is found to be the highest during the acute phase of the disease when MG levels in the respiratory tract are high. The vaccine strain ts-11 is capable of vertical transmission [23].

### 2.2. Horizontal

Horizontal transmission may occur by respiratory aerosols, hatchery transmission, direct contact with infected birds, or indirect modes. Environmental factors and fomites are the reason for transmission through indirect mode. Fomites left in the feeder are the foremost cause of horizontal transmission. Infection via contaminated food is an example of an environmental factor in indirect transmission [24]. The hatchery transmission may occur through debris from broken eggs that act as a source of infection for other birds. Infection transmission studies show that many strains of MG cause lowered egg production, higher hatch failure owing to embryo death, and lowered fertility [25]. A diagrammatic representation of modes of transmission is illustrated in Figure 2. Survival of MG was reported to be longer on several parts of organisms, such as feathers, the contents of eggs, human skin for a day or two, and on bird feeders for one day [26,27]. Regular cleaning and disinfection may reduce the spread of this organism.

## 3. Pathogenesis and Host Immune Response

Different strains of *Mycoplasma* sp. are host specific, but some species have the capability to colonize other species [28]. The flask-shaped structure of the *Mycoplasma* with a terminal tip often exhibits gliding motility, and the tip structure facilitates strong attachment to the host cell. The presence of cytoadherence molecules GapA and CrmA assist in the initial adhesion and attachment of MG to the host cell surface [29]. Subsequent to initial colonization at the upper respiratory tract, MG spreads to the lower respiratory tract and causes bronchitis, airsacculitis, and pneumonia [30]. After primary attachment, MG causes the release of mucus at the epithelial sites as a result of an inflammatory response [29]. MG enters into non-phagocytic cells to escape from the host immune system and spread infection throughout the host [29]. Cytokines released following MG infection play a crucial role in pathogenesis by activation of leukocytes [29]. Infection of MG occurs through the airway and bronchial epithelial cells, followed by attachment to host cell surfaces. The next step involves the inhibition of protein and DNA synthesis in the host cells and reduced mucus production. The systemic invasion of MG into the host cells leads to pathology, immunosuppression, and production losses [28]. A simplified version of the pathogenesis of MG is represented in Figure 3.

The steep increase in antibody titers against virulent proteins of MG indicates the occurrence of a strong humoral immune response in chickens when infected through direct contact and/or aerosols [31]. Systemic IgM was detected in the first week of infection, before infiltration. Later, B-lymphocyte proliferation was noticeable in the trachea, at approximately 3 weeks post-infection [32]. An elevated concentration of IgA antibodies was detected in the trachea after 2 weeks post-*M. gallisepticum* infection [29,33]. These studies conclude that the chicken humoral immune response is crucial in the prevention of infection from MG [29].

The importance of T and B cells against MG infection has been demonstrated. Birds with thymectomy and bursectomy had an extensively increased severity of disease with MG infection compared to control birds [34,35]. The predominant presence of B cells in chicken trachea post-MG infection reflected the crucial role played by the B cells [35]. After 4 days of MG infection, CD4+ T cells and CD8+ T cells were reported to be observed in the lamina propria of the trachea [33]. Cytokines and chemokines also play significant roles at some point in MG infection [36]. B and T cells are non-specifically stimulated by pro-inflammatory cytokines and chemokines. Distorted expression of cytokine and chemokine-related genes such as *CCL-20*, *IL-8*, *IL-12*, *IFN-γ*, *IL-1β*, *MIP-1β*, *RANTES*, *CXCL-13*, and *CXCL-14* were revealed arbitrarily via the activation of toll-like receptor-2 (TLR) by MG lipoproteins. Distorted expression of these genes was also observed in an NF-κB pathway-dependent manner by MG lipoproteins [37]. The above studies imply that subsequent to initial attachment and propagation, cytokines and chemokines mediate a lympho-proliferative inflammatory response in the respiratory tract. Collectively, MG is able to induce both innate and adaptive mechanisms of the host immune responses. Macrophages are important components aiding in phagocytosis and clearance of MG. In this immunological event, B and T cells play critical roles in the clearance and dissemination of MG. Therefore, their diagnosis seems very important in the process of vaccination.

## 4. Diagnosis of *Mycoplasma gallisepticum*

The isolation of the organism in a cell-free medium or direct detection of its DNA in infected tissues or swab samples are the two methods used to confirm the diagnosis of MG infection. The use of serological testing for diagnosis is also common [21]. As a confirmatory test, hemagglutination-inhibition is performed because nonspecific false agglutination reactions might happen, particularly following the injection of inactivated oil-emulsion vaccines or infection with *M. synoviae.* However, advancements in molecular biology have provided a rapid and sensitive replacement for conventional culture methods, which need specialized methodologies and expensive reagents. Polymerase chain reactions, random amplified polymorphic DNA, arbitrarily primed polymerase chain reactions, and multiplex real-time polymerase chain reactions are some of the molecular diagnostic procedures used to identify *Mycoplasma*. ELISA is used as a primary confirmation test in several countries [37]. Bacterial isolation, in addition to serological and genetic tests, is being used for monitoring MG infection. The ability to differentiate between antibodies generated by spontaneous infection and those evoked by vaccination in vaccinated flocks is a limitation of commonly used serological techniques for diagnosis. In contrast, the culture needed for MG isolation may take at least 21 days and may even be hindered by the other bacteria’s fast growth. Molecular genetic assays such as genomic template stability and multi-locus sequence typing, and sequencing could help to differentiate the three vaccination strains, but it takes time [38,39]. To detect specific avian *Mycoplasma* DNA, real-time PCR and conventional PCR are usually employed instead of culture, but these two PCR techniques can only detect avian *Mycoplasma* species or allow for the simultaneous identification of *M. gallisepticum* and *M. synoviae.*

## 5. Virulence and Immune Evasion Proteins

MG encodes a range of proteins that act as virulent proteins, adhesion proteins, lipoproteins, heat shock proteins, and antigenic variation proteins [28,29]. Table 1 outlines the list of key virulence factors associated with MG infection and pathogenesis. The literature review indicated that each of the virulent factors has shown a specific role in the pathogenesis and immune induction of MG.

### 5.1. Adhesions

MG has a high affinity towards chicken respiratory epithelial cells and attaches to their surface [7]. In the host, the colonization of this pathogen occurs in the respiratory tract through a definite attachment organelle, which is tapered at one end (terminal bleb-like structure) of the organism. Hence, cytoadherence is a crucial step in *Mycoplasma* infection, and adhesion proteins play a vital role in this process [29,46]. MG GapA and CrmA are the most important adhesion proteins present within the bleb structure and are indispensable for successful colonization in host cells [29,47]. MG exhibits an adhesion mechanism similar to the human respiratory pathogen *M. pneumonia* [48]. The adhesion protein P1 in *M. pneumoniae* is analogous to MG GapA [49]. In addition to the primary cytoadherence proteins, fibronectin-binding proteins such as PlpA and Hlp3 [41] and the heparin-binding protein OsmC-like protein could contribute to host cell adhesion and colonization.

### 5.2. Immune Evasion

The immune evasion of MG is regulated by *the vlhA* gene family. This family consists of 43 *vlhA* genes located in five loci [43]. The major function of this gene family is to engender antigenic diversity, which assists in immune evasion during infection [38]. The *VlhA* gene family shows phase variation during the acute phase and immune evasion during the chronic phase of infection [43,50]. The phase variation may occur impulsively or by an immune attack, and is crucial for the survival of MG in host cells [51,52,53]. Various mechanisms for phase variation, such as gene conversion, site-specific recombination, DNA slippage, and reciprocal recombination, are utilized by different species of *Mycoplasma*s [53]. The *vlhA* gene products are speculated to be engaged in the attachment of host apolipoprotein A1 [54,55,56] and red blood cells [57]. Among the other *vlhA* genes, *vlhA* 3.03, 2.02, and 4.07 genes are primarily expressed in the initial phase of infection, whereas *vlhA* 1.04 is expressed in the later stages of infection. The prototype followed by MG to express the dominant *vlhA* gene during the course of infection is stochastic and its mechanism is unknown [50].

### 5.3. Phase Variation by Mycoplasmas

One of the important factors for pathogenesis and chronic infection is prodigious phenotypic variation by *Mycoplasma*s [58,59]. A putative cytoadhesion-related protein (PvpA) is a phase variable protein identified by host immune cells [60]. PvpA is an integral membrane protein without lipids and has a surface-exposed C-terminal, showing size variation within strains. *PvpA* genes of MG show high similarity with P30 and P32 proteins of human MG pathogens variants such as *M. pneumoniae* and *M. genetalium* [44].

### 5.4. Heat Shock Proteins

The heat shock proteins are highly conserved and maintain cellular proteins from heat shock, inflammation, and infection [61]. GroEl is among the heat shock protein family, also known as Hsp60 or Group 1 chaperonin. GroEl is identified as a virulent protein and complements adhesion in MG [45].

## 6. Prevention from *M. gallisepticum*

In the poultry industry, the best prevention approach is to obtain *M. gallisepticum* infection-free birds and fertile eggs. This would sustain the MG-free status of the flocks, and birds will be free of infection, sero-conversion, or disease. These will cut off potential vertical MG or hatchery transmission of MG. If this is not achievable for any reason, producers need to learn to live with MG-positive flocks, and put in various prevention strategies to minimize the infection, spread, sero-conversion, disease, and production declines. Strict biosecurity and routine sanitation are quintessential to avoid the spread of MG infection. A large number of poultry farms within a small geographical area increase the probability of exposure and spread of disease [62]. An effective monitoring system and early diagnosis are needed to prevent or control an MG infection outbreak. Sanitation and hygienic methods should be followed during the artificial insemination of birds to decrease potential vertical transmission. In some countries, a number of poultry producers use antimicrobial agents for the treatment and control of MG infection in flocks [63]. The most commonly used antimicrobials in poultry farms are pleuromultilins, macrolides fluoroquinolones and tetracyclines [64]. The extensive use of antimicrobials can lead to the development of antimicrobial resistance in MG, resulting in futile treatment [65]. The lack of cell walls in MG highly reduces the choice of current antibiotics. Another major concern while using antimicrobials is that they can enter the food chain and may cause side effects in humans [66] and an increase in antibiotic resistance. As an alternative to antibiotics, both inactivated and live MG vaccines have been used for decades now. Though there is no complete immunity in poultry against MG infection, broadly, MG vaccination reduces the *Mycoplasma* load, disease severity, and production losses. With international efforts to minimize the use of antibiotics in livestock, the use of MG vaccines and vaccination programs are preferred in a number of countries.

## 7. Currently Available Vaccines against *M. gallisepticum*

Vaccination against MG was first recommended by Adler and his team and has been used as a control measure for mycoplasmosis in poultry since the 1960s [67]. Both bacterins/inactivated and live attenuated vaccines have been used in the prevention of MG [67]. The list of current commercially available vaccines against MG infection is given in Table 2. Due to an increase in antimicrobial resistance and reduced antibiotic efficacy to control MG infections, the elimination of this disease is frequently coupled with novel and effective vaccines, and improvement of existing vaccines.

Many isolates of MG have been used to produce inactivated vaccines in a number of countries. In the literature, the R strain appears to have been made available for many decades, and has been reported to decrease respiratory symptoms, respiratory tract lesions, egg transmission, and production loss due to MG [67,71]. Conversely, others have reported that the protection they offer against infection and respiratory disease was inconsistent. The foremost benefit of oil emulsion bacterins over live attenuated vaccines is the reduced risk of reversion to virulent forms and the ability to induce high levels of humoral antibodies [72].

### 7.1. Bacterins/Inactivated Vaccine

Many isolates of MG have been used to produce inactivated vaccines in a number of countries. The R strain appears to be available for many decades, and have been reported to decrease respiratory symptoms, respiratory tract lesions, egg transmission and production loss due to MG (Table 2). Conversely, the protection they offer against infection and respiratory disease was inconsistent with R strain. The foremost benefit of oil emulsion bacterins over the live attenuated vaccines is the reduced risk of reversion to virulence form, ability to induce high levels of humoral antibodies and protection against drops in egg production (Table 2).

### 7.2. Live-Attenuated Vaccines

#### 7.2.1. First Generation (1975–2000)

The first live attenuated MG vaccine strain described was the F strain. Many experimental challenge studies have confirmed the efficacy of the F strain vaccine in preventing air sac infection, respiratory illness, loss of egg production, and egg transmission [73], with no effects on egg quality [74]. In layer chickens, vaccination with the F strain prior to the onset of laying prevents the shedding of the vaccine strain into eggs, though they tend to shed and transmit the vaccine strain by contact. Still, the F strain is capable of causing respiratory symptoms in broilers [75] and also in turkeys, causing outbreaks in turkeys in the field [76]. Studies show that the F strain can remain in the upper respiratory tract throughout the life of the chicken [77]. The delivery mode for the F strain vaccine is intranasal instillation or coarse spray and eye drops.

In the early 1990′s, the 6/85 vaccine strain was developed in the USA. It is not virulent in chickens and turkeys, and no bird-to-bird transmissions have been reported. This vaccine strain is evident in the upper respiratory tract for 4 to 8 weeks post-vaccination and does not provoke a measurable serological response [76,77], signifying the limitation of the 6/85 vaccine strain in terms of defense and period of immunity [77].

The ts-11 vaccine strain was developed from an Australian MG isolate through chemical mutagenesis [78], and the mode of administration is by eye drop. This vaccine shows less or no virulence in turkeys and chickens, and MG was not transmitted between birds [78]. It offers protection against MG infection, lowers losses in egg production [77,78], and limits vertical transmission of virulent MG [35]. The ts-11 vaccine strain after administration becomes available in the upper respiratory tract throughout the life of the bird and evokes long-term immunity [35]. The protective immunity induced by the ts-11 vaccine is dose-dependent and shows a variable serological response in vaccinated chickens [79,80]. A few studies showed the potential reversion of ts-11 to a virulent strain in the field and that it acquired the potential for vertical transmission [64].

#### 7.2.2. Second Generation (2000-Present)

With the advancement in technology, many MG vaccine candidates such as GT5 [81], MG 7 [29], K-strain [67], and ts-304 [82] have been researched as substitutes to address the concerns about existing vaccine strains. Vaccine candidates GT5 and MG 7 were derived from virulent strain Rlow. The serial passage of the virulent strain Rlow resulted in the Rhigh strain, and subsequent complementation of Rhigh with the cytoadherence gene *GapA* resulted in the GT5 vaccine strain [7,29,81]. The MG 7 strain was developed by insertion of transposon in the middle of the virulent dihydrolipoamide dehydrogenase (*lpd*) gene [83]. Experimental vaccination studies with both GT5 and MG 7 strains have been shown to protect infected chickens from tracheal lesions and colonization of MG in the trachea [81].

The K-strain is persistent in the upper respiratory tract for about 5 months and demonstrated protection for chickens against tracheal lesions caused by MG [84]. The K-strain offers protection equivalent to commercial F strain and ts-11 vaccines [85]. The ts-304, a variant of the ts-11 strain, with the presence of the *GapA* gene tends to be more protective against virulent MG in lower doses than ts-11 in turkeys and chickens [86]. The protective immunity of ts-304 lasts for at least 57 weeks after a lone inoculation at 3 weeks old [86].

### 7.3. Genetic Engineered Vaccines

In recent times, recombinant vaccines are gaining interest. This vaccine comprises the identification and cloning of immunogenic factors in a suitable expression system [87,88]. Functional transposons in *Mycoplasma*s such as Tn916 and Tn4001 were used in the construction of mutant, gene expression, cellular tagging, and protein functional analysis [89]. To date, two recombinant vaccine candidates have been developed against MG infection viz. GT5 (attenuated and genetically modified *M. gallisepticum* strain) and fowlpox virus-encoding MG genes [81,90]. More recently Zhang et al. [88] used a recombinant adenovirus to express the S1 spike glycoprotein of infectious bronchitis virus (IBV) and the TM-1 protein of MG in HEK293 cells. The recombinant adenovirus retained the parental biological characteristics, successfully expressed target proteins, produced elevated levels of antibodies, and considerably decreased the clinical signs and lesions subsequent to IBV and MG challenge [88].

## 8. Future Vaccines

The development of efficacious vaccines against MG is essential for the future. Most of the current vaccine development strategies are based on single antigens or different antigens, but nonetheless in a single shot. Thus, the new epitope-based vaccines are considered to be an excellent futuristic approach. An antigen contains the epitope, which is the basic unit that is capable of eliciting either a cellular or humoral immune response. A multi-epitopic vaccine is made up of a series of epitopic (antigenic) peptides, thus facilitating the prevention of infection or inducing an immune response. An ideal multi-epitopic vaccine is designed in such a way that it should have epitopes that can elicit cytotoxic and helper T lymphocytes and B cells’ immune response against the targeted microorganism [88]. Multi-epitope-based vaccines are advantageous when compared to conventional vaccines. They are cheaper to develop, do not involve microbial culturing, and can outdo several wet lab procedures, thus saving time. Epitope-based vaccines also decrease the risk associated with the reversal of virulence, unlike the live attenuated strains. Furthermore, the epitopes can be sensibly engineered and optimized to boost their effectiveness in evoking stronger immune responses and have increased chemical stability due to their small size. They offer safety, as they do not involve entire pathogens and are extremely specific and stable. Due to the presence of multiple epitopes, the vaccine candidate can bind multiple HLA alleles at a time and can ensure the desired immune response among a heterogeneous population. Multi-epitopic vaccines have been developed for the following poultry pathogens: Newcastle disease virus, avian influenza A (H7N9) virus [91], and Eimeria parasitic infection [92]. Epitope-based vaccines are gaining attention due to their specificity [93]. Recently, a multi-epitope-based vaccine utilizing an up-to-the-date immuno-informatics approach has been proposed against MG [94,95]. The use of recombinant vaccine technologies can endow a safe and efficient vaccine against MG infections, although further exploration and validation are required.

Currently, most of the vaccine expression system for multi-epitopic vaccine production is based on bacterial, yeast, and mammalian expression systems. These conventional vaccine production systems have several disadvantages. For example, in the bacterial expression system, the problems encountered were difficulty in expressing higher eukaryotic proteins, accumulation of endotoxins, and protease contamination of the host. In the yeast expression system, the main setback is hyper-glycosylation, while in the mammalian expression system, the disadvantages of high costs, slow cell growth, and higher chances of contamination were recorded [95].

For the past two decades, apart from the conventional production platforms and technologies used in manufacturing vaccines, drugs, and other biologics by industries and pharmaceutical companies, research on the plant-based production of vaccines for veterinary diseases has gained attention. This is due to exhilarating prospects such as the possibility of developing an edible vaccine that has the potential to develop safe, effective, stable, and economical prophylactics, vaccines, and medicines for a variety of ailments, including infectious disorders. It can also be produced on a large scale at a low cost and with no chance of contamination. In the case of edible vaccines, there would be no need for a cold chain during transportation and storage. In addition, plants can produce and process eukaryotic vaccine proteins better than other expression systems. The purified plant-produced vaccine can be administered via intramuscular, intranasal, ocular, and oral routes [96,97].

The production of a plant-derived vaccine requires the transfer of the target gene into the plant for protein expression (Figure 4). The protein expression can be achieved by stable or transient expression. When the gene of interest is permanently incorporated into the plant genome through nuclear or plastid integration, it is known as stable expression, whereas in transient expression, the production of the desired protein is achieved without integration of the target gene into the plant cell genome.

An ideal vaccine should be effective as well as affordable, easy to administer to a large population, elicit long-lasting cellular and humoral immunity, be non-pathogenic, contain fewer side effects, and be less likely to contaminate the environment, as it has been observed in recent pandemics [98,99,100]. Hence, in the near future, plant-based vaccines will gain popularity because of the above-listed advantages and due to their cost-effectiveness as they are free from cross-contamination from other animal sources, easily transportable in cold-chain storage, and can be easily grown and expanded in a short span of time. Taking these advantages into consideration, for future vaccine development, plants can be used as biofactories. Hence, prior to making a large investment in a possible product, this should be taken into account.

## 9. Conclusions

MG is one of the most common and important diseases in chickens and turkeys, and infection with this pathogen can lead to a reduction in meat or egg production globally. The existing MG vaccines, including live-attenuated, inactivated (bacterins), and recombinant vaccines, have not provided an acceptable or consistent level of protection against MG infection, disease, or production losses in the past decades. With growing worldwide concerns regarding antibiotic resistance against MG, finding an alternative but effective control for MG infection and losses is highly desirable. Vaccines based on the adhesion and phase variation proteins of MG are the most suitable candidates to prevent and control *M. gallisepticum* infection and to sustain better health, welfare, and production of poultry. Selective epitope-based vaccines can also be a potential candidate. Though increasing research has been undertaken on these potential vaccines, ultimately the vaccine type, safety, effectiveness, functional mechanisms, and immunogenicity should be extensively studied through in vivo studies.

## Figures and Tables

**Figure 1 vaccines-11-00469-f001:**
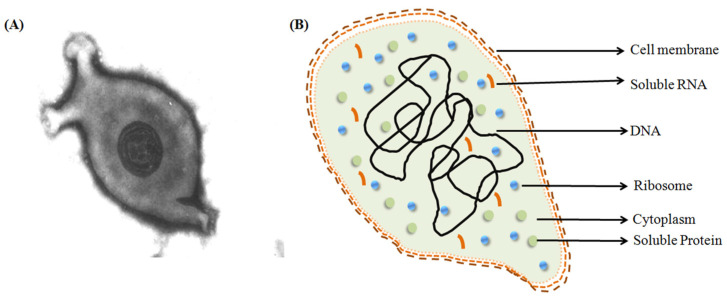
Structure of *M. gallisepticum*. (**A**) Negatively stained image of *M. gallisepticum* (Source: M.H.B. Catroxo and A.M.C.R.P.F. Martins under Creative Commons Attribution 3.0 Unported, https://app.dimensions.ai/details/publication/pub.1004322571, accessed on 12 February 2023 and https://en.wikipedia.org/wiki/Mycoplasma_gallisepticum#/media/File:Mycoplasma_gallisepticum.jpg, accessed on 12 February 2023). (**B**) Representational image of *M. gallisepticum*.

**Figure 2 vaccines-11-00469-f002:**
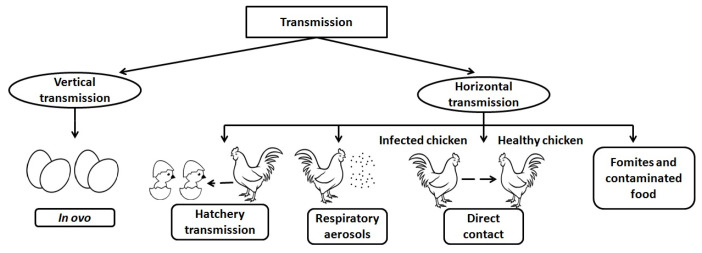
Modes of transmission of *Mycoplasma gallisepticum*.

**Figure 3 vaccines-11-00469-f003:**
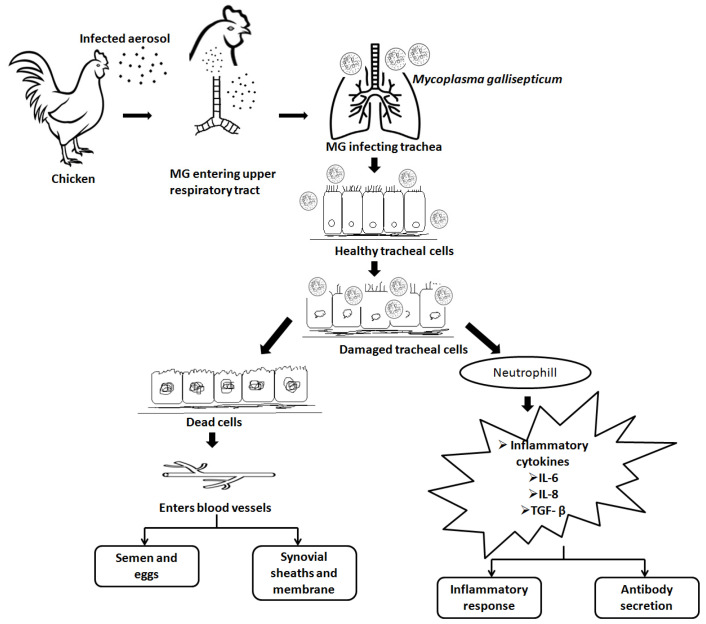
Pathogenesis of *M. gallisepticum*. *M. gallisepticum* enters airways; invades healthy cells; damages healthy cells; systemic invasion, immuno-suppression, health, and production loss.

**Figure 4 vaccines-11-00469-f004:**
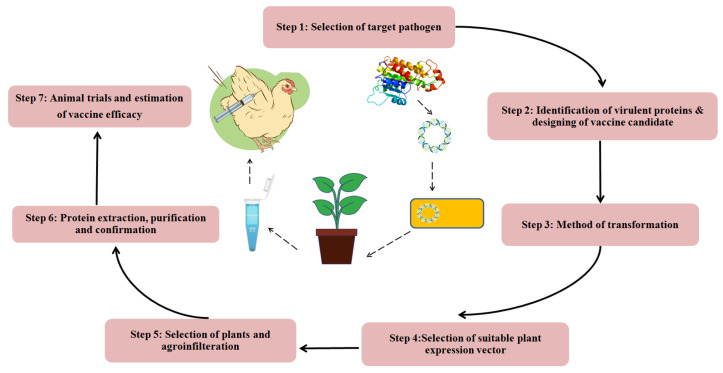
Steps involved in the production of plant-based vaccine.

**Table 1 vaccines-11-00469-t001:** Virulent proteins of *M. gallisepticum* involved in infection process.

Function	Protein	Gene Annotation	Reference
Adhesion	GapA	*M. gallisepticum* A_0934	[29,40]
	CrmA	*M. gallisepticum* A_0939	[29,40]
	Hlp3	*M. gallisepticum* A_0928	[41]
	PlpA	*M. gallisepticum* A_1199	[41]
	OsmC-like protein	*M. gallisepticum* A_1142	[42]
Immune evasion	*vlhA* 1.04	*M. gallisepticum* A_0070	[43]
	*vlhA* 2.02	*M. gallisepticum* A_0117	[43]
	*vlhA* 3.03	*M. gallisepticum* A_0380	[43]
	*vlhA* 4.07	*M. gallisepticum* A_0977	[43]
	*vlhA* 5.13	*M. gallisepticum* A_1261	[43]
Phase variation	PvpA	*M. gallisepticum* A_0258	[44]
Heat shock proteins	GroEl	*M. gallisepticum* A_0152	[45]

**Table 2 vaccines-11-00469-t002:** Examples of commercially available vaccines against *M. gallisepticum*.

Strain	Name	Manufacturer
Strain F	*Mycoplasma gallisepticum* vaccine	Shandong Lvdu Biosciences [68]
	CEVAC MG F	CevaSanteAnimale [68]
	PoulvacMyco F	Zoetis United States [68]
	AviPro^®^ MG-F	Elanco [69]
Strain K	VAXXON^®^ MG Live	Vaxxinova^®^ International BV [70]
Strain ts-11	VAXSAFE MG VACCINE (MG TS-11)	Bioproperties Pty LTD [68]
Strain 6/85	Nobilis MG 6/85	MSD Animal Health [68]
	MYCOVAC-L^®^	Merck [69]
Strain S6	VAXXON^®^ MG Inac	Vaxxinova^®^ International BV [70]
Strain R	MG-Bac Vaccine	Zoetis United States [68]
	AviPro104 MG BACTERIN	Lohmann Animal Health International [68]

## Data Availability

All data generated or analyzed during this study are included in this published article.

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
