# Peer review of "Infection, Transmission, Pathogenesis and Vaccine Development against Mycoplasma gallisepticum"

_vaccines, 2023, doi:10.3390/vaccines11020469_

Round 1

Reviewer 1 Report

Mugunthan et al. reported on the Mycoplasma gallisepticum infection, pathogenesis, and vaccine evelopment. The review is well written. I recommend somecomments.

-        Line 51: Another unique characteristic of mycoplasmas is the 51 use of UGA codon to encode tryptophan, please explain in more detail

-        Mycoplama spp. should be in italics. Please revise throughout the manuscript. e.g., lines 57, 59, 60, 62,…….

-       Lines 70 and 72: please revise the citation style according to the journal instructions.

-        Figure 1: Did the author obtain the Copy right permission?

-        Name of genes e.g., vlhA, should be in italics; please check throughout the manuscript

-        Line 85: Please add the following sentence: MG in chickens is not only causing CRD but also opens the way for other co-infecting pathogens, such as E. coli, and low pathogenic avian influenza subtype H9N1, causing severe economic losses. (Abdelrahman A. A.,  et al. 2021. Avian Mycoplasma gallisepticum and Mycoplasma synoviae: Advances in diagnosis and control. Ger.J. Vet. Res. (2): 46-55).

-        Table 2: very poor table. At least please summarize the main features/disadvantages of each vaccine in a few bullets for each 

-        Please add a chapter to discuss the diagnosis of MG? Diagnosis of MG infections is confirmed by isolation of the organism in a cell-free medium or direct detection of its DNA in infected tissues or swab samples. Serological tests are also widely used for diagnosis (Abdelrahman A. A. et al. 2021. Ger.J. Vet. Res. (2): 46-55).  However, advances in molecular biology represented a rapid and sensitive alternative to the traditional culture methods requiring specialized techniques and sophisticated reagents. Several mycoplasma molecular diagnostic tests are implemented: including polymerase chain reaction (PCR), Random Amplified Polymorphic DNA (RAPD), arbitrary primed polymerase chain reactions (AP-PCR), and Multiplex real-time polymerase chain reaction. Then please discuss in brief these diagnostic techniques.

Author Response

Mugunthan et al. reported on the Mycoplasma gallisepticum infection, pathogenesis, and vaccine development. The review is well written. I recommend some comments.

Reply: We thank the respected reviewer for the encouraging positive comments. We have complied all the comments to revise the ms.

Line 51: Another unique characteristic of mycoplasmas is the 51 use of UGA codon to encode tryptophan, please explain in more detail.

Reply: We have changed the line as “Another unique characteristic of mycoplasmas is the use of UGA codon to encode tryptophan this is due to the presence of tRNA which is capable of reading the UGA termination codon as tryptophan” included in line 51

Mycoplama spp. should be in italics. Please revise throughout the manuscript.

Reply: Amended the correction mentioned by Reviewer.

Lines 70 and 72: please revise the citation style according to the journal instructions.

Reply: Amended the correction mentioned by Reviewer.

Figure 1: Did the author obtain the Copy right permission?

Reply: The image  is licensed under the “Creative Commons Attribution 3.0 Unported” license.

Appropriate credit is provided in a link to the license were given.

Name of genes e.g., vlhA, should be in italics; please check throughout the manuscript.

Reply: Amended the correction mentioned by Reviewer.

Line 85: Please add the following sentence: MG in chickens is not only causing CRD but also opens the way for other co-infecting pathogens, such as E. coli, and low pathogenic avian influenza subtype H9N1, causing severe economic losses. (Abdelrahman A. A.,  et al. 2021. Avian Mycoplasma gallisepticum and Mycoplasma synoviae: Advances in diagnosis and control. Ger.J. Vet. Res. (2): 46-55).

Reply: Amended the correction mentioned by Reviewer in line 85.

Table 2: very poor table. At least please summarize the main features/disadvantages of each vaccine in a few bullets for each 

Reply: Included advantages of each vaccine in Table 2

Please add a chapter to discuss the diagnosis of MG?

Please add a chapter to discuss the diagnosis of MG? Diagnosis of MG infections is confirmed by isolation of the organism in a cell-free medium or direct detection of its DNA in infected tissues or swab samples. Serological tests are also widely used for diagnosis (Abdelrahman A. A. et al. 2021. Ger.J. Vet. Res. (2): 46-55).  However, advances in molecular biology represented a rapid and sensitive alternative to the traditional culture methods requiring specialized techniques and sophisticated reagents. Several mycoplasma molecular diagnostic tests are implemented: including polymerase chain reaction (PCR), Random Amplified Polymorphic DNA (RAPD), arbitrary primed polymerase chain reactions (AP-PCR), and Multiplex real-time polymerase chain reaction. Then please discuss in brief these diagnostic techniques.

Reply: Amended the correction mentioned by Reviewer.

Reviewer 2 Report

Review: Mycoplasma gallisepticum infection, pathogenesis and vaccine development: A Review

General comments:

The subject dealt with by the authors is interesting and topical for the global poultry sector.

The review is complete and contains the most pertinent bibliographic citations, even if not always in large numbers, but this makes reading much easier.

 Abstract:

It is good and clear.

Introduction

It is good and clear.

Line 77-78 says the same thing as line 66, merging the same concept.

 Transmission

It is good and clear.

The diagrams are easy to interpret and very clarifying.

All chapters are written clearly and references are pertinent

 References

In the bibliography the name of the journal must be put in "italic". Please read the instructions on the website https://www.mdpi.com/journal/vaccines/instructions

Author Response

General comments:

The subject dealt with by the authors is interesting and topical for the global poultry sector.

The review is complete and contains the most pertinent bibliographic citations, even if not always in large numbers, but this makes reading much easier.

Reply: We are highly thankful to the respected reviewer for the encouraging and positive comments.

 Abstract:

It is good and clear.

 Reply: Thank you for the comments.

Introduction

Line 77-78 says the same thing as line 66, merging the same concept.

Reply: We have changed the line to avoid repletion.

Transmission

It is good and clear.

The diagrams are easy to interpret and very clarifying.

All chapters are written clearly and references are pertinent

Reply: Thank you for the comments.

References

In the bibliography the name of the journal must be put in "italic". Please read the instructions on the website https://www.mdpi.com/journal/vaccines/instructions

Reply: We have made the corrections mentioned by Reviewer.

Round 2

Reviewer 1 Report

Accept in present form